# Impedance Analysis of a Two-Layer Air-Core Coil for AC Magnetometry Applications

**DOI:** 10.3390/s25175339

**Published:** 2025-08-28

**Authors:** Mateusz Midura, Grzegorz Domański, Damian Wanta, Przemysław Wróblewski, Waldemar T. Smolik, Kamil Lipiński, Michał Wieteska, Piotr Bogorodzki

**Affiliations:** Institute of Radioelectronics and Multimedia Technology, Warsaw University of Technology, 00-655 Warsaw, Poland; mateusz.midura@pw.edu.pl (M.M.); grzegorz.domanski@pw.edu.pl (G.D.); przemyslaw.wroblewski@pw.edu.pl (P.W.); waldemar.smolik@pw.edu.pl (W.T.S.); kamil.lipinski.dokt@pw.edu.pl (K.L.); michal.wieteska@pw.edu.pl (M.W.); piotr.bogorodzki@pw.edu.pl (P.B.)

**Keywords:** multilayer magnetic coil, air core, impedance analysis, magnetic nanoparticles

## Abstract

The aim of the study was to analyze the impedance characteristics of the transmitting coil used in a system for AC magnetometry and the measurement of the Specific Absorption Rate (SAR) of magnetic nanoparticles. A theoretical relationship for the current distribution in a multilayer air-core coil was derived. The formulas for the coil’s stray capacitance were modified to account for additional spacing between layers, introduced to reduce the interlayer capacitance. The developed theory was applied to a two-layer air-core coil with an additional gap between the layers. The frequency dependence of the coil impedance was measured. The measurements confirmed an extension of the useful operating frequency range of the constructed coil.

## 1. Introduction

Magnetic hyperthermia is a modern treatment method that uses a magnetic field to heat magnetic nanoparticles introduced into the body—most often for the purpose of destroying cancer cells [1,2,3,4,5,6]. It is a technique at the intersection of medicine, physics, and nanotechnology. The most commonly used magnetic material is iron oxide (Fe_3_O_4_) or its variants [7]. The nanoparticles can be chemically modified to selectively bind to cancer cells [8]. When exposed to an alternating magnetic field, the nanoparticles, after reaching the tumor site, generate heat and raise the temperature of the tumor, thereby destroying it [9]. One of the key aspects of this technique is measuring the properties of the nanoparticles, particularly the Specific Absorption Rate (SAR) [10]. This can be achieved by placing the nanoparticles in a specially designed excitation coil, whose task is to generate an alternating magnetic field. These measurements are typically conducted within a frequency range from one hundred to several hundreds of kHz [11,12]. The heating efficiency of magnetic nanoparticles depends on many factors, including the frequency of the applied magnetic field, the size of the nanoparticles, their shape, the type of core material, and the coating material [12]. A review of various analytical and numerical models describing the behavior of nanoparticles in an alternating magnetic field is presented in [12]. Another application of excitation coils is the measurement of the properties of magnetic nanoparticles using the Magnetic Particle Spectroscopy method [13,14] and the measurement of the complex magnetic susceptibility using the AC-magnetometry method [15,16]. The impedance of a magnetic coil depends on frequency due to the imaginary component related to inductance, which, assuming constant inductance, increases linearly with frequency, and the real component, associated with the winding resistance, which increases with frequency due to the skin effect. Current flows primarily along the surface of the conductor. As a result, the effective cross-sectional area decreases, causing the resistance to increase with frequency.

Another effect influencing the real part of the impedance is the proximity effect, which refers to the influence of external magnetic fields on the current distribution within the conductor. An additional factor modifying the frequency-dependent behavior of the coil’s impedance is the inter-turn capacitance. As frequency increases, the coil reaches self-resonance. Once the resonant frequency is exceeded, the impedance begins to decrease because the coil starts to behave like a capacitor.

Therefore, to analyze the coil’s impedance as a function of frequency, it is necessary to calculate its inductance, determine the influence of the skin and proximity effects on resistance, and calculate the coil’s stray capacitance.

To increase the magnetic flux density in an air-core coil, a multi-layer winding can be used. However, a disadvantageous factor affecting the range of useful frequencies in magnetic hyperthermia applications is the capacitance between the winding layers. This capacitance can be reduced by introducing air gaps between layers. Another method to extend the useful frequency range of the coil is to use Litz wire for the winding [17,18,19]. Litz wire is composed of numerous individually insulated conducting strands. These strands are twisted, woven, or arranged in hierarchical patterns that combine both twisting and weaving to minimize skin and proximity effects. The use of enamel-coated magnetic wire with specific twist patterns ensures that each strand varies its position relative to the center of the bundle over its length. This variation promotes uniform current distribution among the strands and helps cancel proximity effects. To be effective, each individual strand must have a diameter smaller than the skin depth at the operating frequency.

Equivalent RLC circuits are modeled using lumped elements—such as resistances, self and mutual inductances, and capacitances—derived from analytical methods or Finite Element Method (FEM) simulations [20,21,22,23]. Among computational techniques, FEM is widely used for evaluating capacitive effects, particularly due to its ability to handle complex three-dimensional geometries. Unlike analytical formulas, FEM can account for intricate and anisotropic structures, offering more precise parameter estimation. The accuracy of FEM results improves with the level of detail in the design data provided. Because stray capacitances are difficult to measure directly, numerical methods like FEM serve as valuable tools during the design phase of various devices and instruments [21].

The aim of this study is to analyze the frequency dependence of the impedance of a multilayer air-core coil. For the theoretical analysis of coil impedance, it is necessary to calculate the resistance while accounting for the skin and proximity effects, as well as to calculate the inductance and stray capacitance. In this work, a theoretical expression for the current distribution in a multi-layer air-core coil was derived. The formulas for the coil’s stray capacitance were modified to include additional spacing between layers, introduced to reduce the capacitance between them. A list of symbols and their definitions is given in Table 1.

All components of the electric and magnetic fields in this work are phasors.

## 2. Materials and Methods

The analyzed coil is an infinitely long cylinder with air core and concentric conductive layers with air gap between them. The layer conductors consist of a linear, homogeneous, and isotropic material. The properties of the *i*-th layer are *σ_i_* (conductivity), *μ_i_* (magnetic permeability), and ε_i_ (electric permittivity), and its external radius is *r_i_*, as shown in Figure 1. The harmonic dependence *exp* (*jωt*) of all field quantities is assumed.

The magneto-quasistatic approach (MQS) is assumed in this analysis. At the frequency range of interest of the mentioned applications, the displacement current in the system can be neglected compared to the induced currents in the conductor.

### 2.1. Mathematical Model of Magnetic Field in Multilayer Coil

The circular current $i\left(t\right)=I\cdot cos(\omega t)$ flows perpendicular to the axis of the coil with circular frequency ω=2πf. The cylindrical coordinate system $\left(r,\phi ,z\right)$ is used, in which the axis *z* coincides with the axis of the symmetry of the coil. The length of the coil in *z* direction is *w*. The external radius of the coil is equal to $r_{n}$, where *n* is the number of layers. It is assumed that $w\gg r_{n}$. Then the partial derivates $\frac{\partial }{\partial z}=\frac{\partial }{\partial \phi }=0$, the current density has only the component $J_{\phi }$, the electric field has only the component $E_{\phi }$, and the magnetic field induction has only the component $B_{z}$. The harmonic dependence $e^{j\omega t}$ of all field quantities is assumed. Ampere’s law in differential form leads to the following equation:(1)$$-\frac{\partial B_{z}}{\partial r}=\mu J_{\phi }=\mu \sigma E_{\phi }.$$

Faraday’s law in differential form leads to the following equation:(2)$$\frac{1}{r}\frac{\partial \left(rE_{\phi }\right)}{\partial r}=-j\omega B_{z}.$$
where $j=\sqrt{-1}$ is the imaginary unit. Eliminating electrical field $E_{\phi }$ from the above equations, one can write down:(3)$$\frac{1}{r}\frac{\partial }{\partial r}\left(r\frac{{\partial B}_{z}}{\partial r}\right)+k^{2}B_{z}=0,$$
where the parameter *k* is equal:(4)$$k=\sqrt{-j\omega \sigma \mu }.$$

After substituting x=kr and using $r=\frac{x}{k},\;\frac{\partial }{\partial r}=k\frac{\partial }{\partial x}$, one can obtain:(5)$$\frac{k}{x}k\frac{\partial }{\partial x}\left(\frac{x}{k}k\frac{{\partial B}_{z}}{\partial x}\right)+k^{2}B_{z}=0.$$

If we assume that we are considering a case of a region in which the parameter *k* is nonzero, then we can simplify the equation:(6)$$\frac{1}{x}\frac{\partial }{\partial x}\left(x\frac{{\partial B}_{z}}{\partial x}\right)+B_{z}=0.$$

After differentiating, we obtain the following equation:(7)$$\frac{\partial^{2}B_{z}}{\partial x^{2}}+\frac{1}{x}\frac{{\partial B}_{z}}{\partial x}+B_{z}=0.$$

After multiplying by $x^{2}$, we obtain full form of the Bessel equation, with the order of Bessel function α=0:(8)$$x^{2}\frac{\partial^{2}B_{z}}{\partial x^{2}}+x\frac{{\partial B}_{z}}{\partial x}+\left(x^{2}-\alpha^{2}\right)B_{z}=0.$$

The general solution of Equation (8) in the *i*-th layer is the following:(9)$$B_{z}\left(x\right)=C_{i}J_{0}\left(x\right)+D_{i}Y_{0}\left(x\right),\;r_{i-1}\leq r\leq r_{i},$$
where $J_{0}(x)$ is the Bessel function of the first kind and zeroth order, $Y_{0}(x)$ is the Bessel function of the second kind and zeroth order, and *C_i_* and *D_i_* are constant parameters in the *i*-th layer, dependent on the boundary conditions between the layers. After substituting x=kr and taking into account that in each layer we may have a different value of the parameter *k*, one can obtain the general solution of the Equation (3) in the *i*-th layer consisted of conductor:(10)$$B_{z}\left(r\right)=C_{i}J_{0}\left(k_{i}r\right)+D_{i}Y_{0}\left(k_{i}r\right),\;r_{i-1}\leq r\leq r_{i},$$

The parameter $k_{i}$ is equal:(11)$$k_{i}=\sqrt{-j\omega \sigma_{i}\mu_{i}}.$$

If the *i*-th layer is an air core or air gap, the parameter $k_{i}=0$ and the solution (10) is no longer valid. Equation (3) becomes the following:(12)$$\frac{1}{r}\frac{\partial }{\partial r}\left(r\frac{{\partial B}_{z}}{\partial r}\right)=0,$$
and its general solution with two linear independent functions is the following:(13)$$B_{z}\left(r\right)=C_{i}+D_{i}\ln\left(r\right),\;r_{i-1}\leq r\leq r_{i}.$$

If the index *i* is equal to 1, then we have the air core and the constant $D_{1}=0$, because $\log\left(r\right)\to -\infty$ when r→0. So it is clear that in the air core, the magnetic field is constant. Also, in the middle air gaps, the magnetic field is constant because the current density is zero ($\sigma_{m}=0)$, and then from Equation (1), it is clear that $\frac{\partial B_{z}}{\partial r}=0$. The magnetic field is zero at the external surface of the outermost layer:(14)$$B_{z}\left(r=r_{n}\right)=0.$$

From the above equation, the following equation for the outermost layer immediately follows:(15)$$C_{n}J_{0}\left(k_{n}r_{n}\right)+D_{n}Y_{0}\left(k_{n}r_{n}\right)=0.$$

The *z* component of the magnetic field induction has to be continuous, so one can write down the boundary condition between two conductive layers:(16)$$C_{i}J_{0}\left(k_{i}r_{i}\right)+D_{i}Y_{0}\left(k_{i}r_{i}\right)=C_{i+1}J_{0}\left(k_{i+1}r_{i}\right)+D_{i+1}Y_{0}\left(k_{i+1}r_{i}\right).$$

If the inner layer is an air gap, then the boundary condition is the following:(17)$$C_{i}=C_{i+1}J_{0}\left(k_{i+1}r_{i}\right)+D_{i+1}Y_{0}\left(k_{i+1}r_{i}\right).$$

For a coil containing $N_{l}$ layers of winding, with $N_{m}$ turns in each layer carrying a current *I*, the magnetic field in the air core can be approximated by [24]:(18)$${B_{z}(r)=C}_{1}=\frac{\mu_{0}N_{l}N_{m}I}{w}.$$

For an air gap in *i*-th layer above, which there are $N_{above}$ conducting layers, the magnetic field inside the gap is given by the following formula:(19)$${B_{z}(r)=C}_{i}=\frac{\mu_{0}N_{above}N_{m}I}{w}.$$

The complex amplitude of the current density can be determined according to (1):(20)$$J_{\phi }\left(r\right)=-\frac{1}{\mu_{i}}\frac{\partial B_{z}}{\partial r}=\frac{k_{i}}{\mu_{i}}\left({C_{i}J}_{1}\left(k_{i}r\right)+D_{i}Y_{1}\left(k_{i}r\right)\right).$$

The electric field can be determined for conductive layer:(21)$$E_{\phi }\left(r\right)=-\frac{1}{\mu_{i}\sigma_{i}}\frac{\partial B_{z}}{\partial r}=\frac{k_{i}}{\mu_{i}\sigma_{i}}\left({C_{i}J}_{1}\left(k_{i}r\right)+D_{i}Y_{1}\left(k_{i}r\right)\right).$$

The tangential component of the electric field must be continuous at the boundary, which leads to the following equation at the interface between the *i*-th and (*i*+1)-th conducting layers:(22)$$\frac{k_{i}}{\mu_{i}\sigma_{i}}\left({C_{i}J}_{1}\left(k_{i}r_{i}\right)+D_{i}Y_{1}\left(k_{i}r_{i}\right)\right)=\frac{k_{i+1}}{\mu_{i+1}\sigma_{i+1}}\left({C_{i+1}J}_{1}\left(k_{i+1}r_{i}\right)+D_{i+1}Y_{1}\left(k_{i+1}r_{i}\right)\right).$$

For a multilayer coil, the corresponding system of equations for the coefficients $C_{i}$ and $D_{i}$ is constructed as follows: For the first layer, i.e., the inner air core, $D_{1}=0$, and $C_{1}$ is given by Equation (18). For the next layer, which is a conductor, the condition at ${r=r}_{1}$ is given by Equation (17), while the outer boundary of this second layer is subject to condition (17) for an air gap, or conditions (16) and (22) if the next layer is a conductor. Proceeding in this way, we reach the outermost layer, for which condition (15) applies.

The DC resistance of the solenoid is equal:(23)$$R_{DC}=\frac{2\pi }{w}\sum_{i\in Con}\frac{1}{\sigma_{i}ln\left(\frac{r_{i}}{r_{i-1}}\right)},$$
where the summation is carried out over the indices *i*, for which the layer is conductive. If we approximate a coil, which has $N_{m}$ turns in each layer, using a cylindrical multilayer model, the above formula is modified to the form:(24)$$R_{DC}=\frac{2\pi }{w}\sum_{i\in Con}\frac{N_{m}^{2}}{\sigma_{i}ln\left(\frac{r_{i}}{r_{i-1}}\right)},$$

The average power dissipated in the *i*-th conductive layer is equal to [19]:(25)$$P_{i}=\frac{\pi w}{\sigma_{i}}\int_{r_{i-1}}^{r_{i}}{\left|J_{\phi }(r)\right|}^{2}rdr.$$

On the other hand, the average power is related to the active resistance of the *i*-th layer by the following expression:(26)$$P_{i}=\frac{1}{2}R_{i}I^{2}.$$

From both equations above, the active resistance of the *i*-th layer can be determined, and then summarized to calculate the total resistance of the coil:(27)$$R=\sum_{i\in Con}R_{i}=\sum_{i\in Con}\frac{{2P}_{i}}{I^{2}}.$$

The above theory was used to determine the real component of the coil’s impedance. The developed coil contained an air core and two conductive layers separated by an air gap. Therefore, the considered coil model consists of a total of four layers: the air core, the first (inner) conductive layer, the air gap, and the second (outer) conductive layer; see Figure 2.

The system of equations for the unknown coefficients in the air layer is as follows:(28)$$C_{1}=\frac{{2\mu }_{0}N_{m}I}{w},\;D_{1}=0,C_{3}=\frac{\mu_{0}N_{m}I}{w},\;D_{3}=0\;.$$

The above expressions are valid for the air core and the air gap. The analogous expressions for the first (inner) conductive layer are the following:(29)$$C_{2}J_{0}\left(k_{2}r_{1}\right)+D_{2}Y_{0}\left(k_{2}r_{1}\right)=C_{1},\;C_{2}J_{0}\left(k_{2}r_{2}\right)+D_{2}Y_{0}\left(k_{2}r_{2}\right)=C_{3}.$$

The expression for the second (outer) conductive layer is the following:(30)$$C_{4}J_{0}\left(k_{4}r_{3}\right)+D_{4}Y_{0}\left(k_{4}r_{3}\right)=C_{3},\;C_{4}J_{0}\left(k_{4}r_{4}\right)+D_{4}Y_{0}\left(k_{4}r_{4}\right)=0.$$

After solving the system of Equations (29) and (30), we obtain the desired coefficients $C_{2},D_{2},C_{4},D_{4}$. The magnetic field distribution can then be calculated using Equation (10), while the current distribution can be determined using Equation (20). The inductance *L* of the two-layer coil can be calculated using the magnetic field distribution $B_{z}(r)$ inside the coil, its outer radius $r_{4}$, the current *I*, and the number of turns in each layer *N_l_*, using the formula:(31)$$L=Re\left[\frac{2\pi N_{l}}{I}\int_{0}^{r_{2}}B_{z}\left(r\right)rdr+\frac{2\pi N_{l}}{I}\int_{0}^{r_{4}}B_{z}\left(r\right)rdr\right].$$

After solving the system to calculate the series resistance in a winding made with Litz wire, the theory presented in [18] was used. The ratio of AC resistance to DC resistance is calculated using the formula [18]:(32)$$K_{d}=\frac{R_{AC}}{R_{DC}}=\frac{\zeta }{\sqrt{2}}\left(\psi_{1}\left(\zeta \right)-\frac{\pi^{2}N_{o}\beta }{24}\left(16N_{m}^{2}-1+\frac{24}{\pi^{2}}\right)\psi_{2}\left(\zeta \right)\right),$$
where $N_{0}$—number of strands in a conductor, β—packing factor, ζ—parameter calculated from the equation [18]:(33)$$\zeta =d_{s}\sqrt{\pi f\mu_{0}\sigma },$$
where $d_{s}$—strand diameter. The functions $\psi_{1},\;\psi_{2}$ are given by the following approximate expressions [18]:(34)$$\psi_{1}\left(\zeta \right)=2\sqrt{2}\left(\frac{1}{\zeta }+\frac{\zeta^{3}}{3\cdot 2^{8}}-\frac{\zeta^{5}}{3\cdot 2^{14}}\right),\;\psi_{2}\left(\zeta \right)=\frac{1}{\sqrt{2}}\left(-\frac{\zeta^{3}}{2^{5}}+\frac{\zeta^{7}}{2^{12}}\right)$$

Normalization was used for the calculations—the $K_{d}$ factor was calculated according to the formula above and then divided by the $K_{d}$ factor computed for a frequency close to zero. To fit the theoretical resistance curves to the measured ones, an effective strand diameter equal to $\eta d_{s}$ was used, where η is a correction factor determined experimentally.

### 2.2. Model of Stray Capacitance in Multilayer Coil

Another factor affecting the coil’s impedance is the stray (or distributed) capacitance. In a multilayer coil, it consists of the capacitance between turns within the same layer and the capacitance between turns in adjacent layers. This self-capacitance arises from interactions between adjacent current loops [25]. It is primarily determined by two components: the turn-to-turn capacitance $C_{tt}$ between neighboring turns within the same layer, and the layer-to-layer capacitance $C_{ll}$ resulting from adjacent turns in different layers. Model of orthogonal winding coil with independent pitch in the axial and radial directions is shown in Figure 3.

The diameter of the winding wire is denoted as $d_{0c}$, while the outer diameter, including the enamel, is $d_{0}$. The radial spacing between the centers of the wires in successive layers is $p_{r}$, and the spacing between the centers of the wires in the same layer, i.e., in the Z-axis direction, is $p_{z}$. The permittivity of vacuum is denoted as *ε*_0_, and the relative permittivity of the enamel (or other wire insulation) is *εᵣ*. The length of one turn is denoted as $l_{t}$. The outer and inner diameters of the air core coil are represented as *D* and *d*, respectively. For a capacitor composed of two parallel shells, the elementary capacitance of the cylindrical coating ${dC}_{in}$ is given by the following Equation [25]:(35)$${dC}_{in}=\frac{\varepsilon_{r}\varepsilon_{0}l_{t}}{ln\left(\frac{d_{0}}{d_{0c}}\right)}d\theta .$$

Two such infinitesimal capacitances appear in series along with the capacitance of the air gap between the turns of the winding wire. This third infinitesimal capacitance, associated with the air gap, is given by the formula from [25], with introducing additional gap between wires in Z direction:(36)$${dC}_{gap}=\frac{\varepsilon_{0}l_{t}d_{0}}{2\left(p_{z}-d_{0}cos\theta \right)}d\theta =\frac{\varepsilon_{0}l_{t}}{2\left(\frac{p_{z}}{d_{0}}-cos\theta \right)}d\theta .$$

Two infinitesimal insulation capacitances, together with the air gap capacitance, connected in series, result in an infinitesimal capacitance between the turns for an infinitesimal angle dθ. The total infinitesimal turn-to-turn capacitance is given by the following equation:(37)$${dC}_{tt}=\frac{{dC}_{in}{dC}_{gap}}{{dC}_{in}+{2dC}_{gap}}d\theta =\frac{\varepsilon_{0}l_{t}d\theta }{2\left(\frac{1}{\varepsilon_{r}}ln\left(\frac{d_{0}}{d_{0c}}\right)+\frac{p_{z}}{d_{0}}-cos\theta \right)}.$$

Let us introduce the parameter *a*, given by the formula:(38)$$a=\frac{1}{\varepsilon_{r}}ln\left(\frac{d_{0}}{d_{0c}}\right)+\frac{p_{z}}{d_{0}},\;a>1.$$

To calculate the capacitance between the layers in the formula above, the parameter $p_{z}$ must be replaced with $p_{r}$. Then the turn-to-turn capacitance is determined by the integral:(39)$$C_{tt}=\int_{\theta_{1}}^{\theta_{2}}\frac{\varepsilon_{0}l_{t}d\theta }{2\left(a-cos\theta \right)}=\frac{\varepsilon_{0}l_{t}}{2}\left(F\left(\theta_{2}\right)-F(\theta_{1})\right),$$
where the function *F* is given by:(40)$$F\left(\theta \right)=\frac{2}{\sqrt{a^{2}-1}}arctan\left(\sqrt{\frac{a+1}{a-1}}\tan\left(\frac{\theta }{2}\right)\right).$$

For calculating the capacitance *C_tt_* for a conductor in one of the middle layers, based on symmetry and assuming equal spacing in the radial and axial directions, the integration limits can be taken from −π/4 to +π/4 [25]. For different spacings in the radial and axial directions, the following integration limits integration in the form of angles *θ*_1_ and *θ*_2_ can be adopted:(41)$$\theta_{1}=-arctan\left(\frac{p_{r}}{p_{z}}\right),\;\theta_{2}=arctan\left(\frac{p_{r}}{p_{z}}\right).$$

Similarly, for calculating the capacitance between layers *C_ll_*, also based on symmetry, for equal spacing in the radial and axial directions, the integration limits can be taken from −π/4 to +π/4 [25], whereas for different spacings in both of these directions, the following integration limits can be adopted:(42)$$\theta_{1}=-arctan\left(\frac{p_{z}}{p_{r}}\right),\;\theta_{2}=arctan\left(\frac{p_{z}}{p_{r}}\right).$$

The total capacitance of multilayer coil is the combination of all of the turn-to-turn and layer-to-layer capacitances. For standard winding, the total capacitance is given by the expression [25]:(43)$$C=\frac{1}{N^{2}}\left[C_{tt}\left(N_{m}-1\right)N_{l}+C_{ll}\frac{N_{m}\left(4N_{m}^{2}-1\right)}{3}\left(N_{l}-1\right)\right],$$
where $N=N_{l}N_{m}$. The capacitance theory was used to estimate resonant frequency of a coil. Equivalent electrical circuit of magnetic coil is shown in Figure 4.

The impedance of the equivalent circuit is given by:(44)$$Z\left(\omega \right)=\frac{R}{{\left(1-\omega^{2}LC\right)}^{2}+{\left(\omega RC\right)}^{2}}+j\omega \frac{L\left(1-\omega^{2}LC\right)-RC^{2}}{{\left(1-\omega^{2}LC\right)}^{2}+{\left(\omega RC\right)}^{2}}.$$

Resonance frequency is equal:(45)$$f_{res}=\frac{1}{2\pi \sqrt{LC}}.$$

Knowing *f_res_* and inductance *L* one can compute capacitance *C*.

### 2.3. Measurement of the Series Resistance of the Excitation Coil

The developed hybrid system for the magnetic characterization of superparamagnetic nanoparticles consists of an excitation coil and two sensing probes [26]. The excitation coil has been designed as a double-layer coil with 122 turns, wound at a distance of 200 mm, with 61 turns in each layer. Figure 5 shows the schematic diagram of the system used to measure the series resistance of the excitation coil.

The developed coil has two winding layers, separated by 3.5 mm. The inner diameter of the inner coil former, on which the first layer was wound, was 30 mm. The outer diameter of the inner coil former was 48 mm.

To characterize the impedance and resistance of the coil, a three-node measurement setup was used based on the Analog Discovery Pro ADP3450 device manufactured by Analog Devices, One Analog Way, Wilmington, MA, USA. The waveform generator (W1) applies a sinusoidal excitation directly to the coil, while the voltage across the coil is measured by Oscilloscope Channel 1 (C1). The other terminal of the coil is connected in series with a 100 mΩ reference resistor (1% tolerance), across which the voltage drop is measured using Oscilloscope Channel 2 (C2). All grounds (W1, C1, C2, and resistor) are tied together. This configuration (W1–C1–coil–C2–R–GND), illustrated in Figure 6, is recommended by Analog Discovery documentation for impedance measurements of components that are normally loaded at the output. The assembled measurement system, along with the software, enables the measurement of the dependencies of impedance modulus, series, and parallel resistance, as well as series and parallel reactance.

## 3. Results

Three air-core excitation magnetic coils were designed and built. The first was a single-layer coil, but it was found to generate a magnetic field of insufficient strength. Therefore, a decision was made to design and construct a two-layer coil. Impedance measurements as a function of frequency showed that the coil reached its self-resonance at a frequency that was too low within the intended operating range for magnetic nanoparticles used in hyperthermia (from 100 kHz to 800 kHz [12]). To increase the resonance frequency of the two-layer coil, a third coil was built featuring two winding layers separated by an additional 3.5 mm gap. Measured and calculated parameters of the examined coils are given in Table 2.

The stray capacitances *C*, presented in Table 2, were determined based on the measured resonant frequencies and the measured inductances of the coils. Theoretical and measured dependence of the single-layer coil’s resistance on frequency is shown in Figure 7. Using the theoretical formulas (32)–(34) for AC resistance, the dependence of the ratio of AC to DC resistance on frequency was calculated for the designed coils. The theoretical resistance–frequency characteristics were calculated for a packing factor β = 0.5785 and an additional factor η = 0.8, correcting the effective strand diameter in the Litz wire to *η·dₛ*. The radii are: $r_{1}=24\;mm,\;r_{2}=27.3\;mm,\;r_{3}=30.8\;mm,\;r_{4}=34.1\;mm$.

With increasing frequency, in the considered range up to 800 kHz [12], an increase in the AC resistance of the coil is observed. The theoretical and measured frequency dependence of the resistance of a two-layer coil without additional spacing is shown in Figure 8. A clear resonance is visible at a frequency of about 350 kHz, which is nearly in the middle of the intended operating frequency range of the excitation coil.

To shift the resonant frequency upward, so that it clearly exceeds 800 kHz, a spacing of 3.5 mm was introduced between the two layers. The theoretical and measured frequency dependence of the resistance of the two-layer coil with additional spacing is shown in Figure 9. In the intended operating range of the excitation coil, no resonance is visible.

The theoretical dependence with a correction factor of 0.8 shows good agreement with the measurement results. One of the factors affecting the resonant frequency of the coil is its stray (parasitic) capacitance. For the adopted model of a two-layer coil, it was decided to reduce this capacitance by introducing a gap between the layers. Using the theoretical relationships (38)–(42), the dependence of the coil’s stray capacitance on the additional spacing between the two winding layers was calculated. The results of the calculations are shown in Figure 10.

A decrease in capacitance is observed as the distance between the winding layers increases. For the calculations of the dependence of the coil’s stray capacitance on the additional spacing, the following parameter values were assumed: *d*_0_ = 3.3 mm, *d*_0*c*_ = 3.1 mm, εᵣ = 3, *lₜ* = 0.2177 m. For zero spacing, the stray capacitance was found to be 1016 pF, whereas for a spacing of 3.5 mm, it was 32.2 pF. These calculation results are in qualitative agreement with the measurement results presented in Table 2 for the two-layer coils.

## 4. Discussion

The paper presents an analytical theory describing the distribution of the magnetic field and current in a multilayer air-core coil with air gaps between the conducting layers. The proposed theory applies to coils whose length is significantly greater than their diameter. In the developed model, the configuration and order of the layers can be arbitrary—for example, two conducting layers may be in direct contact with each other, without any air gap between them.

For the adopted model of a two-layer coil, theoretical distributions of the magnetic field and current density were calculated. The calculated frequency dependence of resistance increases significantly faster than the measured resistance of the actual coil. This discrepancy is due to the fact that the coil is wound with Litz wire. To match the theoretical results to the measurements, a known theory of series resistance for Litz wires from the literature was employed. Good agreement between theory and experiment was achieved only after introducing a correction factor for the effective strand diameter in the Litz wire.

The results indicate that introducing a controlled interlayer gap can effectively shift the resonant frequency beyond the target operating band, improving coil stability at higher frequencies. For optimal design, parameters such as packing factor, strand diameter, and dielectric spacing should be selected to balance low stray capacitance with minimal winding resistance. Litz wire significantly reduces skin and proximity effects at higher frequencies, improving efficiency, but increases manufacturing complexity, cost, and winding density constraints.

The paper also extends the theory of stray capacitance known from the literature by considering arbitrary additional spacing between the coil windings—both in the longitudinal and radial directions. A theoretical dependence of the stray capacitance on the thickness of the additional air layer was calculated. When compared to the measured resonance frequency, it turned out that the theoretically determined capacitance was too low. This is likely due to the fact that the coil was wound with Litz wire, for which the theory developed for standard insulated wire is insufficient. In such a case, satisfactory results are likely to be obtained only using numerical methods based on the finite element method (FEM). Therefore, the next stage of work should involve the development of a more accurate theory of stray capacitance for coils wound with Litz wire. Combining finite element method simulations with analytical capacitance and resistance models could enable more accurate, frequency-dependent predictions while reducing computation time, supporting rapid design iterations for high-frequency coil applications.

## 5. Conclusions

The analytical theory developed here can be used to analyze multilayer coils wound with conducting foil and with arbitrary radial structure. Such a theory, enabling the modeling of long air-core coils with air gaps between winding layers, can be used to develop a hybrid method combining the analytical approach with the finite element method. This hybrid method for calculating the electrical and magnetic parameters of the coil could be both accurate and fast, supporting the rapid design of coils with the desired properties.

## Figures and Tables

**Figure 1 sensors-25-05339-f001:**
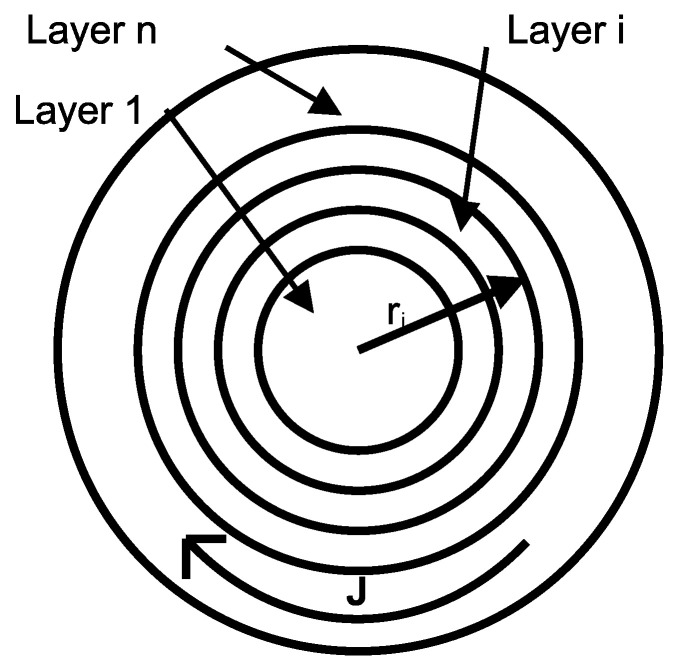
The geometry of a multilayer cylindrical conductor coil featuring an air core and interlayer air gaps.

**Figure 2 sensors-25-05339-f002:**
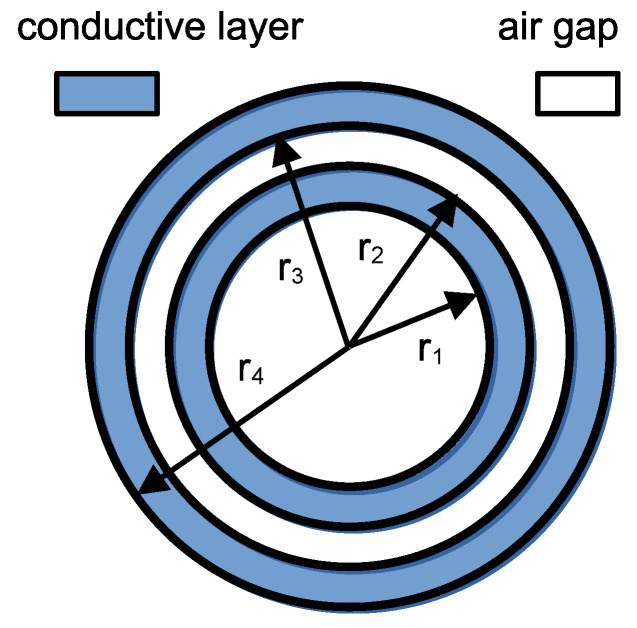
The geometry of the developed two conductive layer magnetic coil featuring an air core and interlayer air gap.

**Figure 3 sensors-25-05339-f003:**
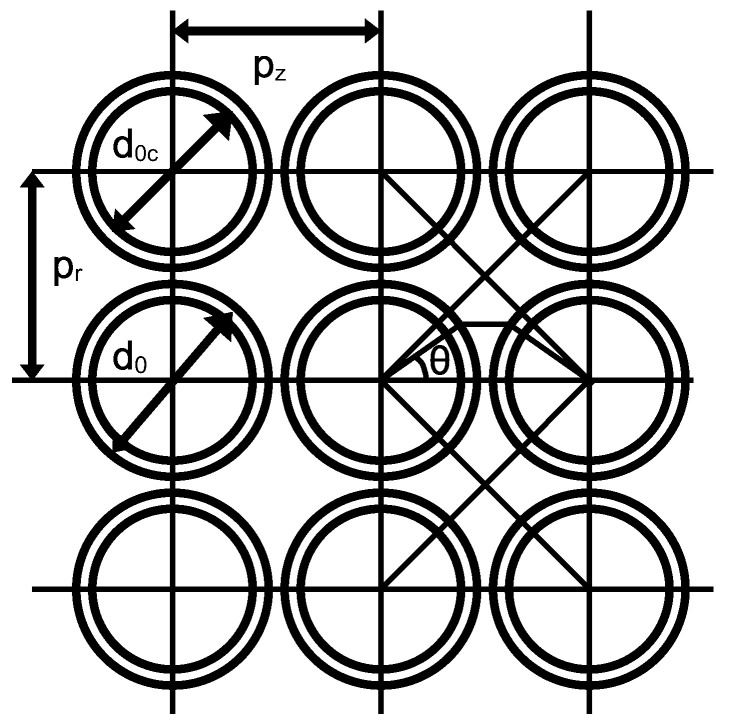
Model of an orthogonal winding coil with independent pitch in the axial and radial directions.

**Figure 4 sensors-25-05339-f004:**
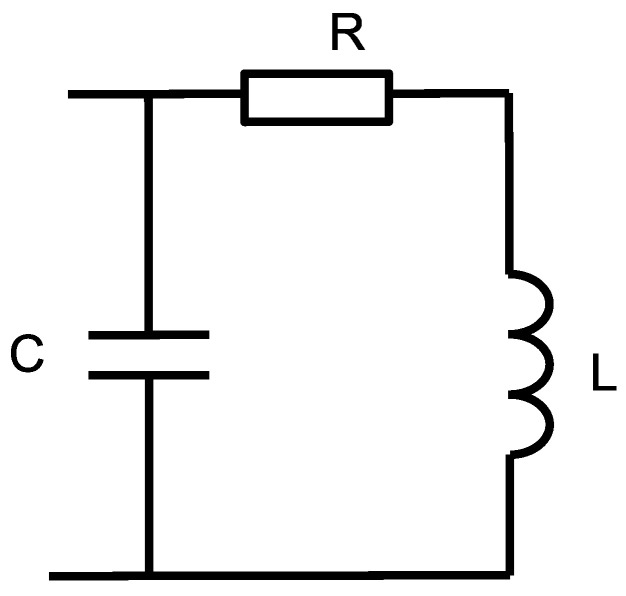
Equivalent electrical circuit of magnetic coil.

**Figure 5 sensors-25-05339-f005:**
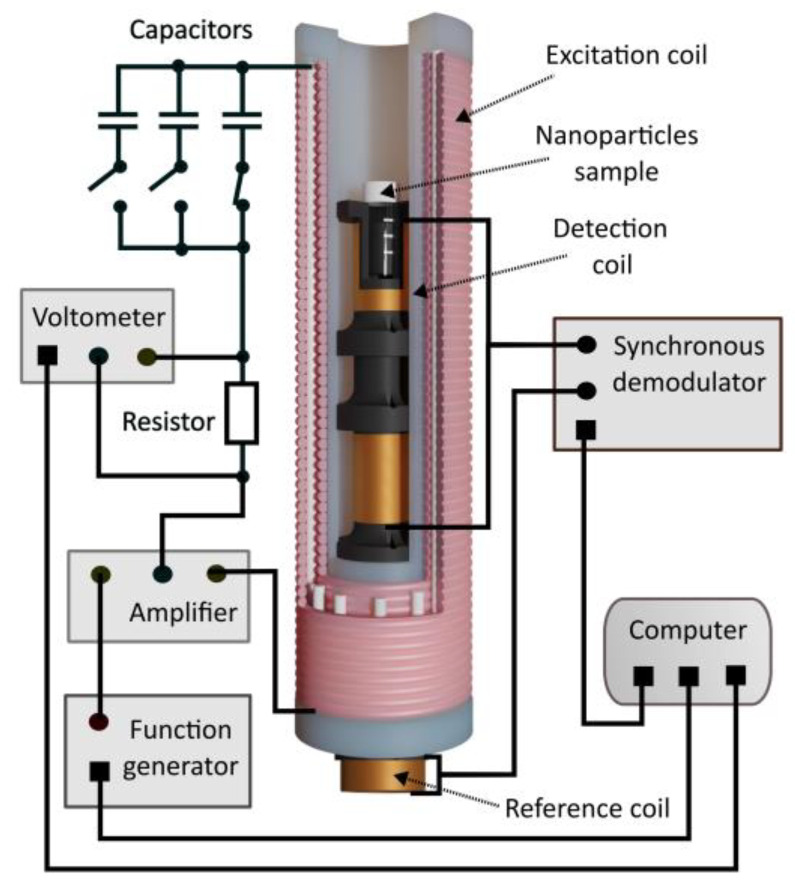
The schematic diagram of the system used to measure the series resistance of the excitation coil.

**Figure 6 sensors-25-05339-f006:**
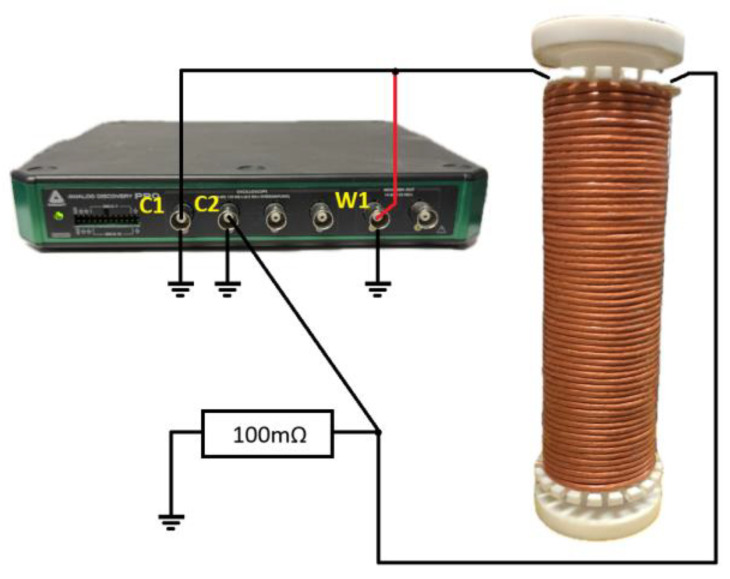
Measurement setup for coil impedance and resistance characterization using the Analog Discovery Pro ADP3450. W1—waveform generator output; C1, C2—oscilloscope input channels of the same device; 100 mΩ reference resistor (1% tolerance).

**Figure 7 sensors-25-05339-f007:**
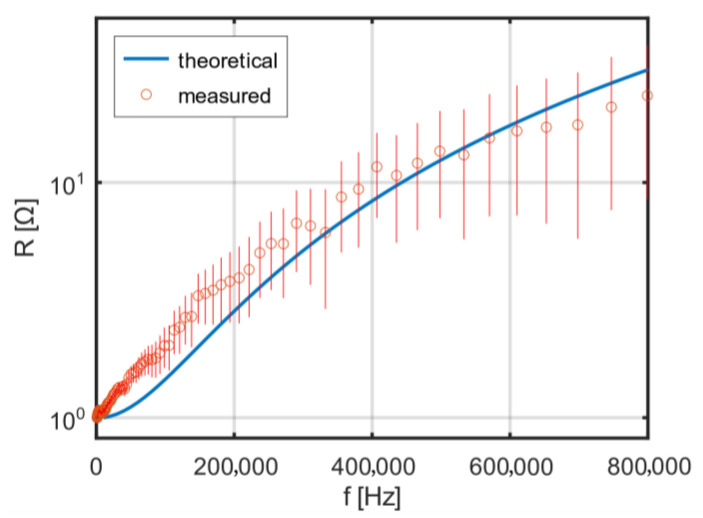
Theoretical and measured dependence of the single-layer coil’s resistance on frequency.

**Figure 8 sensors-25-05339-f008:**
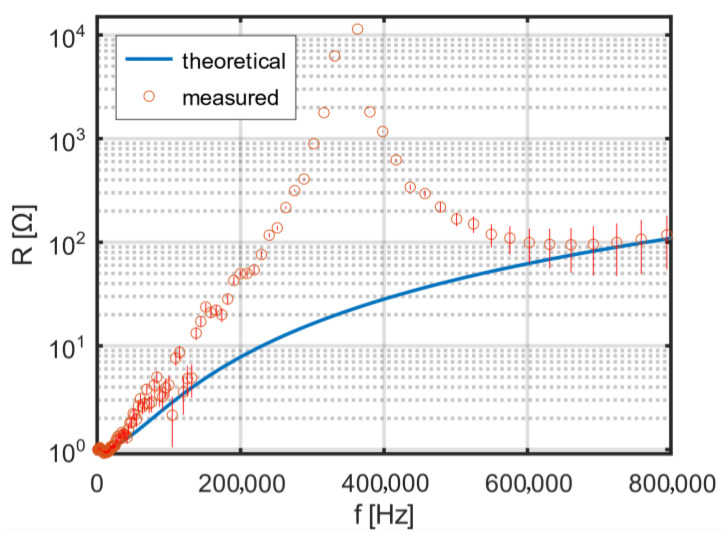
Theoretical and measured frequency dependence of the resistance of a two-layer coil without an interlayer gap.

**Figure 9 sensors-25-05339-f009:**
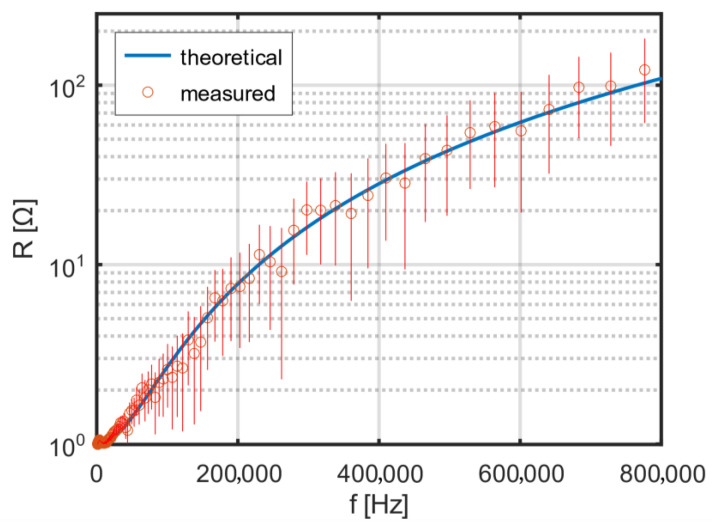
Theoretical and measured frequency dependence of the resistance of a two-layer coil with an interlayer gap.

**Figure 10 sensors-25-05339-f010:**
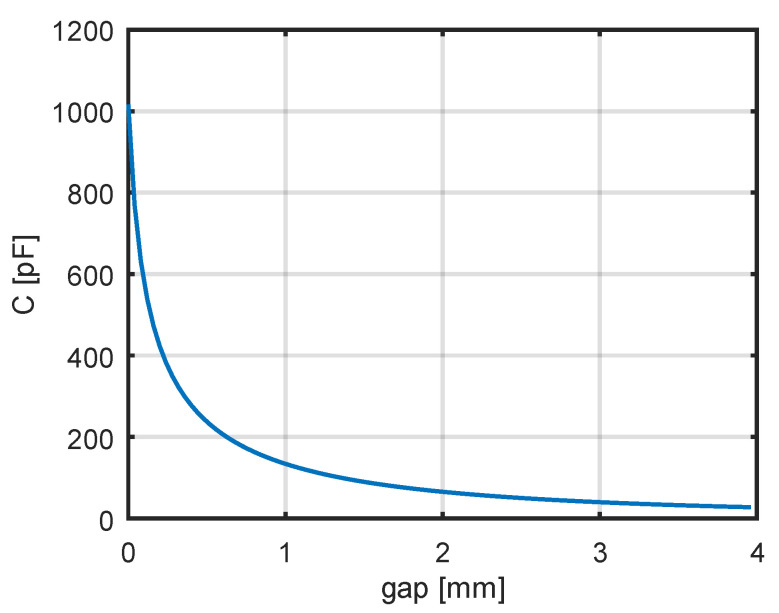
The theoretical dependence of coil capacitance on the gap between layers.

**Table 1 sensors-25-05339-t001:** List of symbols and their definitions.

Symbol	Definition	Unit
*a*	parameter in the integral	dimensionless
*B_r_*	radial component of magnetic induction	T
*B_z_*	axial component of magnetic induction	T
*B_ϕ_*	angular component of magnetic induction	T
*C_i_*	coefficient in magnetic field equation	T
*C_ll_*	layer-to-layer capacitance	F
*C*	total capacitance of coil	F
*C_tt_*	turn-to-turn capacitance	F
*d*	inner diameter of coil	m
*D*	outer diameter of coil	m
*D_i_*	coefficient in magnetic field equation	T
*d* _0_	wire outer diameter	m
*d* _0*c*_	wire core diameter	m
*d_s_*	strand diameter	m
*E_ϕ_*	angular component of electric field	V/m
*f_res_*	resonance frequency	Hz
*i*	index of the layer	dimensionless
*i_c_*	current	A
*I*	current amplitude	A
*J* _0_	Bessel function of the first kind, order zero	dimensionless
*J* _1_	Bessel function of the first kind, order one	dimensionless
*J_ϕ_*	angular component of current density	A/m^2^
*K_D_*	ratio of AC resistance to DC one	dimensionless
*L*	inductance of the coil	H
*l_t_*	turn length	m
*k_i_*	wavenumber in *i*-th layer	1/m
*N*	total number of turns	dimensionless
*N_l_*	number of turns in each layer	dimensionless
*N_m_*	number of layers	dimensionless
*N* _0_	number of strands in Litz wire	dimensionless
*P_i_*	power loss in *i*-th layer	W
*p_r_*	pitch in radial direction	m
*p_z_*	pitch in axial direction	m
*r*	radial coordinate	m
*R*	total resistance of the coil	Ω
*R_AC_*	AC resistance	Ω
*R_DC_*	DC resistance	Ω
*t*	time	s
*w*	length of the magnetic coil	m
*Y* _0_	Bessel function of the second kind, order zero	dimensionless
*Y* _1_	Bessel function of the second kind, order one	dimensionless
*z*	axial coordinate	m
*Z*	coil impedance	Ω
*β*	packing factor	dimensionless
*ε* _0_	electric permittivity of vacuum	F/m
*ε_r_*	relative electric permittivity	dimensionless
*θ*	planar angle	rad
*μ_i_*	magnetic permeability of *i*-th layer	H/m
*μ* _0_	magnetic permeability of vacuum	H/m
*σ_i_*	conductivity of *i*-th layer	S/m
*ϕ*	azimuthal angle	rad
*ω*	angular frequency	rad/s

**Table 2 sensors-25-05339-t002:** Measured and calculated parameters of the examined coils.

Parameter	Single Layer Coil	Two Layer Coil Without Gap	Two Layer Coil with Gap
L measured	56 ± 1 μH	180 ± 1 μH	173 ± 1 μH
L calculated	47 μH	213 μH	229 μH
f_re*s*_	6.10 ± 0.10 MHz	0.35 ± 0.01 MHz	1.74 ± 0.02 MHz
*C*	12.2 ± 0.5 pF	1149 ± 66 pF	48.3 ± 1.1 pF

## Data Availability

The original contributions presented in this study are included in the article. Further inquiries can be directed to the corresponding author.

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
