# Peer review of "Impedance Analysis of a Two-Layer Air-Core Coil for AC Magnetometry Applications"

_sensors, 2025, doi:10.3390/s25175339_

Round 1
Reviewer 1 Report
Comments and Suggestions for Authors
The research presents a theoretical and experimental analysis of impedance characteristics in multilayer air-core coils, particularly in the context of magnetic hyperthermia and nanoparticle SAR measurement applications. This research also provided the analytical understanding of multilayer coil impedance, particularly for air-core designs with interlayer gaps. The topic in this manuscript is relevant to the field of sensors, since this research extends the theory of stray capacitance known from the literature considering arbitrary additional spacing between the coil windings — both in the longitudinal and radial directions. The authors developed theory which can be applied to a two-layer air-core coil with an additional gap between the layers. However, a few questions need to be clarified before it can be considered for publication.
- Real-world coils (e.g., short solenoids) may exhibit edge effects, potentially altering impedance predictions. The authors need to justify this.
- The authors only tested a two-layer design with fixed geometry (3.5 mm gap, 30 mm diameter). Why do the authors think this is enough for the conclusion?
- The authors focus on a limited frequency range (100–800 kHz), but omit higher frequencies (e.g., >1 MHz) where Litz wire effects dominate. Does it cause problems for getting a sturdy conclusion?
- The authors used an assumption of infinite-length of coils, with simplifying boundary conditions, an explanation should be provided for this assumption, since the simplified boundary conditions may affect its generalizability.
- The authors may compare their analytical model with FEM simulations (e.g., COMSOL) to quantify errors in Litz wire capacitance.
- The conclusion is too simple, the authors should extend their conclusion section. "However, for coils wound with Litz wire, this theory is inadequate. In such cases, satisfactory results are likely to be obtained only using numerical methods based on the finite element method (FEM)", this sentence can be in discussion section, thus only one sentence is left in the conclusion part "The analytical theory developed here can be used to analyze multilayer coils wound with conducting foil and with arbitrary radial structure", anyway, the authors should make their conclusion in more detail.
Author Response
Dear Reviewer,
Thank you very much for your thoughtful and detailed review of our manuscript. We truly appreciate the time and effort you invested in evaluating our work. Your insightful comments and constructive suggestions have been extremely helpful in identifying areas for improvement and strengthening the overall quality of the paper.
Please see the attachment.
Sincerely,
On behalf of all authors,
Corresponding Author

Reviewer 2 Report
Comments and Suggestions for Authors
This is a well-organized and technically rigorous paper that presents a theoretical and experimental analysis of the impedance of multilayer air-core magnetic coils, with specific attention to applications in magnetic hyperthermia and nanoparticle characterization. The novelty lies in modifying the stray capacitance model to account for interlayer gaps, and in adapting existing theory to Litz wire behavior. However, there are some issues that need to be solved before publication
Major issues
While the theoretical sections are rigorous, the notation is dense and could be more digestible: 1) Include a table of symbols used throughout the paper. 2) Label all figures and equations with explanatory captions or contextual interpretation in the main text. 3) Add more intuitive explanation between steps, especially when switching from general electromagnetic laws to Bessel-function-based solutions.
The experimental setup is only briefly described. More detail on how measurements were taken (e.g., calibration, test frequency sweep, temperature control) would improve reproducibility. Moreover,iInclude error bars or uncertainty estimates in Figures 5–7 to show confidence in measured values.
The discussion mostly focuses on matching experimental data to theory via correction factors. It would benefit from more insights into: 1)Practical implications (e.g., optimal coil design recommendations) 2)Trade-offs in using Litz wire vs. other techniques (e.g., reducing proximity effect vs. increasing complexity). 3)Future improvements (e.g., hybrid FEM + analytical models).
Authors should also refer to works that have presented analytical and numerical estimations of SAR ( see https://doi.org/10.1039/D5NA00258C) in the introduction part. They should also refer to works that have employed FEM in coil simulation and optimization in the conclusion part.
Minor issues
Typographical: Line breaks in author affiliations and figure captions should be cleaned up.
Language: Mostly clear, but could benefit from minor polishing. For example:
Replace “the coil may exhibit self-resonance” with “the coil reaches self-resonance.”
Use consistent tense in describing figures (present or past).
Figures: Ensure Figure 6 is properly labeled—it appears misattributed ("theoretical dependence of current density on the radius") when it shows resistance vs. frequency.
Author Response
Dear Reviewer,
Thank you very much for your thoughtful and detailed review of our manuscript. We truly appreciate the time and effort you invested in evaluating our work. We are grateful for your valuable feedback, which we believe will significantly contribute to enhancing the clarity and impact of our research.
Please see the attachment.
Sincerely,
On behalf of all authors,
Corresponding Author

Round 2
Reviewer 2 Report
Comments and Suggestions for Authors
The authors revised manuscript according to the proposed comments. The manuscript can be published in the present form.